# Perceptions of women and their partners on postabortion intrauterine contraception: A qualitative study in central Uganda

Herbert Kayiga[1]*, Emelie Looft-Trägårdh[2], Othman Kakaire[1], Nazarius Mbona Tumwesigye[3], Musa Sekikubo[1], Joseph Rujumba[4], Josaphat Byamugisha[1], Kristina Gemzell-Danielsson[2]

**1** Department of Obstetrics and Gynecology, Makerere University College of Health Sciences, Kampala, Uganda, **2** Department of Women's and Children's Health, Karolinska Institutet, and WHO Collaborating Centre, Karolinska University Hospital, Stockholm, Sweden, **3** Makerere University School of Public Health, Makerere University College of Health Sciences, Kampala, Uganda, **4** Department of Paediatrics and Child Health, Makerere University College of Health Sciences, Kampala, Uganda

\* hkayiga@gmail.com

## Abstract

### Background

The uptake of intrauterine devices (IUDs) has stalled below two percent among married Ugandan women. We explored the perceptions of women and their partners on the utilization of postabortion IUDs after medical management of incomplete abortions in central Uganda.

### Methods

Between August 2022 and May 2023, using a semi-structured interview guide, fifteen in-depth interviews were conducted among Ugandan women and their partners at five public facilities on their perceptions in regards to postabortion IUDs. Using inductive content analysis, themes and subthemes were generated.

### Results

Three themes emerged: 1) perceived women's and their partners' barriers in accessing postabortion IUDs such as myths and misconceptions on IUDs, spouse refusal, IUD related side effects. 2) women's and their partners' experiences while using postabortion IUDs such as increased lubrication, freedom from prior contraceptive side effects, assurance of early return to fertility after IUD removal, menstrual irregularities and abdominal pain following IUD insertion. 3) motivators and recommendations to the uptake of IUDs such as peer influence, client-healthcare provider relationship, spousal approval of IUDs and community sensitization on IUDs using social media platforms.

**Data availability statement:** All data underlying the study's findings, have been availed in the manuscript. In case any more data or materials are needed, they are readily accessible from the corresponding author on request.

**Funding:** This work was funded by the Swedish Research Council (external to my organization), (Grant 2019-04256) received by KGD, Makerere University (internal) and the MakRif Project (internal) received by HK in form of tuition waivers and research funds. There was no other additional external funding received for this study. The content is solely the responsibility of the authors and does not necessarily represent the official views of the Swedish Research Council, Makerere University or the MakRif Project. The funders provided support in the form of tuition waiver and research expenses, but did not have any additional role in the study design, data collection and analysis, decision to publish, or preparation of the manuscript.

**Competing interests:** The authors have declared that no competing interests exist.

**Abbreviations:** DMPA, Injectable Depo-Medroxyprogesterone Acetate; IUD, Intrauterine Device; LNG-IUD, Levonorgestrel Intrauterine Device; PAC, Post Abortion Care; RCT, Randomized controlled clinical trial; SOMREC, The School of Medicine Research and Ethics Committee; UNCST, Uganda National Council for Science and Technology; WHO, World Health Organization

## Conclusion

Understanding the socio-cultural context of women and their partners, is pivotal in the uptake of postabortion IUDs. Healthcare providers ought to provide evidence-based counselling to demystify individual and community misconceptions on IUD use. Male partner involvement, the assurance of early return to fertility after IUD removal, user champions, and social media platforms can enhance the uptake of postabortion IUDs.

## Introduction

Unintended pregnancies, which are either unplanned or mistimed pregnancies [1], continue to be prevalent globally yet with dire economic, health and social consequences [2]. The highest global rates of unintended pregnancies are reported in Eastern and Middle Africa [1]. In Uganda, more than half of all pregnancies are unintended, contributing to nearly 1.2 million births [3]. This rate is higher than the global rate of intended pregnancies of 41% [1] and the 33.9% in the sub-Saharan Africa [4]. With restrictive abortion laws in Uganda, women and their partners struggle with the immediate consequences of the unintended pregnancies [5]. The high unintended pregnancy rate may contribute to high rates of induced abortions that might ultimately end up as unsafe abortions when conducted by unskilled personnel or in unsafe environment [6]. In Uganda, about 54 per 1000 women of childbearing age undergo unsafe abortions, of which two out of ten of such women die from abortion-related complications [6–8]. Twenty-five percent of pregnancies end up as spontaneous abortions in Uganda [3]. As a fundamental State's obligation to provide gender equality, and sexual reproductive human rights for sustainable development [9], there is a need to provide effective contraception in Uganda. As fertility can return within two weeks of medical or surgical evacuation [10], effective contraception can prevent adverse pregnancy outcomes such as recurrent pregnancy losses, intrauterine growth restriction and perinatal mortality in the subsequent pregnancies [11,12].

Despite proven effectiveness, safety, and failure rate of less than one percent of intrauterine contraception [13], uptake among Ugandan women has stagnated below two percent among married and one percent among sexually active unmarried women [14] for reasons that may not be well documented. The available intrauterine devices (IUDs) in Uganda are Copper T 380 A and Levonorgestrel IUDs, both deemed safe for postabortion use [15].

Women's and their partners' perceptions on contraception have been explored globally to understand their influence on the uptake of IUDs in prior studies [16–19]. Some studies conducted in sub-Saharan Africa have noted that misconceptions on the complications of IUDs deter women from IUD use for fears of infertility [17], cancer [20,21] and weight changes [17,22,23]. Menstrual irregularities associated with IUDs have also been reported to discourage their uptake [24]. A systematic review in sub-Saharan Africa showed that spousal disapproval of contraception is a major hindrance to utilization of modern contraception [18]. Lack of appropriate knowledge

on contraception among women and their partners, has been reported in Ghana [25] and Nigeria [26] to limit the uptake of the contraception. In Nigeria, women perceived that the more children they bore, the less likely that their husbands could marry other wives [27]. Such a perception turned out to be a hindrance to the uptake of contraception. Socio-cultural influences, such as religious beliefs in South Africa and Ethiopia [28,29], community power dynamics that perceive use of contraception as an 'act of infidelity' [30], have also been reported as barriers to contraception. Women's age [31–33], marital status [34–36] socio-economic and education background [37] have also been reported to influence the decisions to utilize intrauterine contraception. Whether these perceptions on modern contraception are the hindrance to uptake of postabortion IUDs in central Uganda remains unclear.

Prior women's experiences on IUD use generally in Uganda [38], showed that though women knew about IUDs, less than 40% of them had the correct information. More than half of the women were not aware of the availability of the method in the neighboring health facility. Nearly 60% of the women believed that IUDs reduce sexual pleasure, damage their wombs and cause cancer [38]. With most of the prior studies interviewing women and men separately on uptake of contraception [16–19], there is uncertainty on whether exploring the women and their partners' perceptions jointly would generate context-based information that would enhance the uptake and utilization of postabortion IUDs. With this contextual background, this study set out to explore women's and their partners' perceptions on postabortion intrauterine contraception in central Uganda, with the aim of understanding the barriers and motivators to the uptake and use of IUDs.

## Materials and methods

### Study design

Using the case study design [39], between 1st August 2022 and 31st May 2023, fifteen in-depth interviews were conducted in this qualitative study. This qualitative study was nested in a broader randomized controlled clinical trial (RCT) on postabortion contraception after medical management of first trimester incomplete abortions in central Uganda (*forthcoming*). To better understand the barriers and motivators of postabortion IUDs, women included in the RCT and their partners were invited to participate in this study.

### Study setting

Using purposive sampling with maximal variation, fifteen women and their partners aged between 18 and 49 years were invited to participate in the study at five public facilities in central Uganda. The five facilities were purposively selected because of their high patient volume with regards to postabortion care and contraceptive counselling (Appendix 1). At the five health facilities postabortion care services, that include emergency abortion treatment and counselling, contraceptive methods counselling and provision, linkage with other reproductive health services, and community-service provider partnerships [40], were readily available. With provision of postabortion IUDs fully paid for in the trial, we sought to explore the perceptions of women and their partners towards postabortion IUDs, in regards to the socio-cultural, interpersonal, community influences and power dynamics using the conventional inductive content analysis approach [41,42]. To be eligible, the women and their partners had to have opted to use postabortion IUDs, and had to be physically accessible for the in-depth interviews. Women who were unwilling to have their partners involved, were excluded.

### Data collection procedure

**Study materials.** We used a semi-structured interview guide consisting of open-ended questions, to collect information on participants' socio-demographic characteristics, and the various the barriers and motivators in the utilization of the IUDs. The interview questions aimed to also capture experiences while using the postabortion IUDs. The guide also had questions tailored on the relevance of contraceptive counselling and its impact on the decision making, barriers and facilitators to using the IUDs, whether peers, healthcare providers, religion or community, had any influence on the

decision to use the IUDs. Any recommendations the women and their partners had to encourage other women and their partners to use the IUDs, were also explored in the guide. The questions were translated into Luganda. We ensured back-to-back translation (English to Luganda) of the data collection tools to minimize on misinterpretation of the data collected.

**Data collection.** Fifteen in-depth interviews were conducted in either English or Luganda, the commonly spoken local language in central Uganda, by two interviewers and two note takers with prior experience in qualitative inquiry. The two interviewers and note takers were trained prior to data collection. Lead research nurses at the five health facilities helped to purposively sample women and their partners in the RCT based on the women's and their partners' knowledge and experience with postabortion IUDs. We aimed at interviewing women and their partners at various points within the six months of utilizing the IUDs. We also purposively sampled women and their partners of different educational backgrounds and residence from urban, semi-urban and rural areas. The women's and their partners' socio-demographic details such as phone contacts, places of residence and next of kin, were retrieved from the databases. Phone calls or text messages were used to invite potential participants for the interviews. Information about the study was shared in a translated informed consent highlighting the study objective, selection criteria, risks and benefits involved while participating in the study and confidentiality commitments.

Once a verbal consent was offered by the potential participants, a time for the interview for the women and their partners was fixed based on the women's and their partners' convenience thereafter. After obtaining written consent, we conducted joint women's and their partners' interviews in quiet rooms free from interference at the health facilities. The interviews lasted between 30–60 minutes. The first author 'HK' met with the research team for debriefing on the questions, concerns, key emerging themes that arose after the interviews. The interviews were audio-recorded, transcribed verbatim and translated into English. The authenticity of the translated interview content was validated by someone not part of the study 'AS'. Data collection and analysis were conducted concurrently to identify any emerging themes or concepts. The research team ensured that the data collected was thorough and comprehensive. After twelve interviews, three more interviews were conducted and there were no more emerging themes, concepts or insights that were arising from the data. There was redundancy in findings from the data collected. Since there was theoretical saturation, with no further modification or expansion of the theory, we assumed data saturation and stopped the interviews. All transcripts were pseudonymized and were kept in a secure location accessible only to the study team.

**Quality control.** We observed and documented the non-verbal communication expressed by the women and their partners throughout the interviews to understand the power dynamics influencing the women's decisions to take up and use postabortion IUDs. The interview guide was piloted among three women and their partners who opted for postabortion intrauterine contraception at a separate health facility. These women and their partners were purposively sampled and the data collected was used to modify four questions in the guide. The modifications in the interview guide were made to better capture the study objectives and adjust the language appropriately.

## Data analysis

We used the conventional inductive content analysis approach to analyze the data [41–43]. Rigor was ensured in the study according to the Lincoln–Guba criteria [44]. 'RM' and 'HK' listened to the interviews back and forth thereafter transcribing them word-for-word, and phrase-for-phrase. After the transcription of the interviews, 'RM', 'SA' and 'HK' read through the notes and ensured that there were no disparities between the audio-recordings and the transcription notes. The transcripts were then imported into the NVivo 12 plus software for management and analysis. 'RM', 'SA' and 'HK' thereafter generated meaning units, which were later condensed and organized into codes. Key concepts and themes arising from the data were organized into codes through an iterative process of reading the interview transcripts for recurring patterns. From the transcripts, initial codes were identified that were later on transformed, labelled and a codebook generated. Similar codes were subsequently organized into subcategories and categories. Themes were generated from the categories based on their repetitiveness in terms of similarities or differences. RM', 'SA' and 'HK' thereafter reviewed

and defined the final themes that emerged. This iterative process was continuously ongoing in throughout the study. In cases of disagreements, the research team comprising three members had to discuss until a consensus was reached.

### Framework analysis

We used the 1995 version of the Andersen's Behavioural Model of Health Services [45] to explore the perceived barriers and motivators towards the uptake of postabortion IUDs by women and their partners in central Uganda. The model was selected as it was ideal in predicting the factors that influence the utilization of postabortion IUDs by women and their partners as has been used in prior studies [46–48]. This multilevel Behavioral model incorporates both the individual and contextual determinants of health utilization, underpinning the objective of our study [49]. Throughout the data collection, analysis and interpretation of the results, the predisposing factors that we explored included; marital status [50], prior contraceptive experience, age [31–33], women's and their partners' education status [51], and proximity to the health facilities [52]. Family economic status was considered as a potential barrier as women and their partners with financial constraints though interested in using the IUDs, were thought to be unable to access the contraception [53]. Myths and misconceptions within the women and their partners and the community, fear of IUD side effects, cultural beliefs and religion, were hypothesized as barriers to the uptake of IUDs in the current study [54]. Among the enabling factors explored included; partner support, communication within the family, number of current and desired children, assurance of early return of fertility and free access to postabortion IUDs [55]. The perceived needs explored included women's and their partners' desire to prevent adverse pregnancy outcomes [56], the postponing of the subsequent pregnancies due to the economic pressures and desire to be healthier.

### Ethical considerations

Women and their partners fluent in either English or Luganda, the commonest local language in central Uganda, were invited to participate in the study. As mandated by the Institutional review board, women and their partners were compensated for their time. Eligible women and their partners, gave their written informed consent prior to enrolment. Participants were reassured that participating in the study was voluntary and that they could opt out of the study at any time without compromising the relationship with the research team and service delivery. Ethical approvals were obtained from the Makerere University School of Medicine Research and Ethics Committee, (Mak-SOMREC-2021-131), and Uganda National Council for Science and Technology (HS2111ES). Administrative clearances were also obtained from the five implementing health facilities.

## Results

Between August 2022 and May 2023, fifteen women and their partners between 18–49 years were interviewed from five health facilities (Table 1).

From the analysis, three themes emerged. Theme one covered the women's and their partners' perceived barriers to the utilization of the postabortion IUDs. Theme two covered the users' experiences while using the IUDs. The third theme covered the perceived motivators towards the use of postabortion IUDs. From the results interpretation using the Andersen's Behavioral Model of Health Services [45], the predisposing factors towards the utilization of the IUDs included, the marital status, woman's and her partner's education background, prior contraceptive experience. Among the barriers mentioned included; fear of IUD side effects, myths and misconceptions at community and individual level about IUDs, transportation challenges in accessing the health facilities, spousal refusal to take on IUDs. The enabling or motivating factors included; spousal approval of the IUDs, free access to IUD services, and the good client-healthcare provider relationships. From the need factors identified, participants noted that the improved sexual life and freedom from the prior contraceptive complications and peer influence enabled women and their partners to utilize postabortion IUDs. The desire to avoid subsequent adverse pregnancy outcomes and postponement of the next pregnancies were noted as the perceived needs for women and their partners in the utilization of postabortion IUDs.

**Table 1. Socio-demographic characteristics of the 15 Women and their Partners.**

| Characteristic | Overall N = 30 n (%) |
|---|---|
| **Gender** | |
| Males | 15 (50.0) |
| Females | 15 (50.0) |
| **Age** | |
| Below 24 | 2 (6.7) |
| 25-29 | 9 (30.0) |
| 30-34 | 4 (13.3) |
| 35-39 | 6 (20.0) |
| 40-44 | 3 (10.0) |
| 45-49 | 3 (10.0) |
| 50+ | 3 (10.0) |
| **Education** | |
| No formal education | 2 (6.7) |
| Primary | 8 (26.7) |
| Secondary | 10 (33.3) |
| Diploma | 5 (16.7) |
| Degree | 3 (10.0) |
| Masters' degree | 2 (6.7) |
| **Religion** | |
| Catholic | 8 (26.7) |
| Protestant | 9 (30.0) |
| Pentecostal | 7 (23.3) |
| Jehovah's witness | 2 (6.7) |
| Muslim | 4 (13.3) |
| **Occupation** | |
| Peasant | 10 (33.3) |
| Teacher | 4 (13.3) |
| Business | 8 (26.7) |
| Motorcyclists | 5 (16.7) |
| Unemployed | 3 (10.0) |
| **Parity** | |
| No children | 4 (13.3) |
| 1-2 | 7 (23.3) |
| 3-4 | 8 (26.7) |
| 5-6 | 7 (23.3) |
| 7+ | 4 (13.3) |

n stands for absolute frequency and % stands for Row percentage.

### Findings: 'Perceived barriers, user experiences and perceived motivators towards postabortion IUDs'

Theme one covered the perceived women's and their partners' barriers to accessing the postabortion IUDs. This theme encompassed misconceptions and myths that were spread on IUDs, community perceptions on the IUDs, socio-cultural norms and beliefs on the utilization of IUDs, and spousal refusal to use IUDs. There was a transport barrier that made it

hard for the women to access family planning services, fear of IUD side effects, and annoying God when the women and their partners used the IUDs.

The second theme covered the women's and their partners' experiences while using the IUDs. Positive experiences reported included increased sexual lubrications, reduced side effects like weight gain when compared to prior contraceptive methods, the assurance on fertility return immediately the IUDs were removed, and the family ties that grew stronger once the women and their partners agreed and executed the plan of using their desired contraception. The verbalized negative experiences included menstrual irregularities associated with the IUDs, and family separations once a partner learnt that the wife was using an IUD without his consent and the abdominal pains associated with the IUD use.

The third theme covered the perceived motivators to the postabortion IUD use and women's and their partners' recommendations to improve the uptake of the postabortion IUDs. Some of the motivating factors that were mentioned included; the bond that was created with the healthcare providers in the process of counselling and insertion of the IUDs, the feeling of involvement and commitment that came with having men involved in the decision to utilize postabortion IUDs. Recommendations included the provision of holistic counselling by the healthcare providers, having peer champions to reach out to the women and their partners in the community, and community sensitization on IUD use to demystify the prevailing misconceptions and myths (Table 2).

### Perceived Women and Partners' Barriers to the utilization of postabortion IUDs

**Influence of cultural norms, beliefs, religion and personal perceptions on IUD utilization.** In central Uganda, cultural norms and beliefs influence the choices on utilization of IUDs. Some women and their partners believed that IUDs were ill-intended and were meant to limit the growth of the African race. Religion, though believed not to be unanimously against postabortion IUDs, in opting to use the IUDs, it was perceived by some women and their partners as disobedience to God. God commanded the human race to go, multiply and fill the earth. Spousal refusal to use IUDs, transport limitations, myths and misconceptions were reported to be a hindrance to the utilization of IUDs.

**Table 2. Summary of the themes, categories and meaning units.**

| Theme | Category | Meaning units |
|---|---|---|
| 1.Perceived barriers towards the IUD uptake and use | 1.Influence of cultural norms and beliefs on IUD use. 2.Religion and its impact on choice to use IUDs. 3.Personal and interpersonal factors restricting IUD use. | 1.Cultural misbeliefs that IUDs make women infertile. 2.Myths that IUDs can cause cancer and deaths. 3.Religious misbeliefs that using contraception annoys God. 4.The insertion procedure is invasive. 5.Couples lack transport to come for IUDs. 6.Husbands refuse their partners to use IUDs. |
| 2.User experiences with the postabortion IUDs | 1.IUDs improve sexual life. 2.IUDs altered menses and caused sexual inconvenience | 1.Better sexual life with increment in sexual fluids. 2.Easy return to fertility after IUD removal. 3.IUDs have lower side effect profile as compared to other methods. 4.Reduced pelvic infections with IUDs. 1.Menstrual irregularities associated with IUDs. 2.Husbands complain of pricks from IUD strings. 3.IUDs cause abdominal pains especially during menses. |
| 3.Motivators and Recommendations towards uptake of postabortion IUDs | 1.Healthcare providers' support pivotal in IUD uptake. 2.Involvement of partners stirs IUD utilization. 1.User champions, and Community sensitization on post abortion IUDs are key in postabortion IUD use. | 1.Whatever the healthcare providers recommend as contraception is what we shall use. 2.As long as we agree, she can use the method. 1.Assurance of the healthcare providers' support in face of any IUD challenges. 2.Peer influence on IUD use is pivotal in the uptake of IUDs 3.The community needs regular updates on IUDs over social media platforms. |

**Cultural beliefs on IUDs.** Whereas some women and their partners believed cultural beliefs did not influence their decision to take on postabortion IUDs, others strongly believed that culture still has a pivotal role in the uptake of family planning methods. Some cultural leaders like kings and chiefs were reported according to the women and their partners to still have a role in whether to encourage their subordinates or sabotage the efforts to use family planning methods. Some leaders were reported to emphasize that the larger the population, the wider the market base and the better the survival of their communities. There was also a belief that family planning methods were from the west and were meant to wipe out the African race. Such misbeliefs went a long way in discouraging women and their partners from using the postabortion IUDs.

*"Yes, culture is so important because when the IUD had just been inserted, I told my clan leader. He told me that the IUD is not good. It (IUD) can stop a woman from having periods. Those devices are meant to kill us Africans and make you barren. As time has gone by, we experienced no challenges. I confirmed IUDs are good".* **Couple 2-BKY-230331**

Some of the women and their partners disagreed with the notion that their local leaders had any significant influence on their utilization of postabortion IUDs.

*"It is impossible, the king is not the one who decides for me. I decide for myself because I am grown up. No, all tribes use the contraceptive methods. I see my sisters are all using family planning, that is why I also allowed to use the IUD".* **Couple 1-MTN-20230331**

**Religious beliefs against contraception.** The participants articulated different perceptions in regards to the impact of religion on their contraceptive choices. Some of the women and their partners believed that the command "go, multiply and fill the earth" was in the olden days when there were very few people. With the current economic pressures, there was a need to regulate the number of children a woman and her partner could have to avail them the basic necessities of life. Religion therefore wasn't a hindrance to such women and their partners. Some women and their partners felt that they annoyed God having opted to have IUDs yet he wanted them to have as many children as possible. Such women and their partners felt that they needed to consult their religious leaders on the next steps once the IUDs were removed.

*"But also, religion is key in deciding whether we should use the IUDs or not. There is no verse in the Bible that talks about family planning. It says produce and multiply. I think this is because there were bushes with no people and we needed to fill the space. The beautiful fact is that no one can see that we are using an IUD. So, no one can judge us unless we tell them".* **Couple 2-BKY-230331**

**Personal factors.** *IUDs perceived to cause infertility and cancer*: Some of the women and their partners reported a number of myths and misconceptions that had kept them from using the IUDs. Women reported that they feared that once the IUDs were inserted, they would seal off their uteruses. As a result, they would be barren or the IUDs would cut their babies in the subsequent pregnancies. IUDs were also believed to cause cervical cancer. This fear was worsened when they heard that at times even the menses would disappear while using the IUDs especially the LNG-IUDs. Most of the women and their partners also believed that IUDs would be pushed out of their positions during sex. The displacement could at times send the IUDs to the heart and cause death or uterine rupture.

*"…they say that when women use IUDs, they can become barren.I feared that I could not get pregnant again if it is inserted because I had never used family planning before. The healthcare provider explained that it (IUD) has no effect on fertility".* **Couple 2-BKY-230331**

*"I used to have fear before but I was counselled when I came here. Some people used to say that it (IUD) can disappear and goes out of the uterus. It may move up to the heart and you die. I also feared that it (IUD) could disappear in me during sex. I thought he might push it (IUD) deep inside me".***Couple 1-MYN-16022023**

***IUD insertion procedures are intrusive***: Even when some women and their partners acknowledged that the IUDs were effective, the fear of what the procedure entailed discouraged some women from taking on this method. They perceived the procedure to encroach on their privacy. Some women acknowledged that the insertion procedure, made some of their peers to opt for other contraceptive methods that didn't warrant undressing before strangers.

*"I'm naturally a very shy woman. The fact that the healthcare provider was going to look at my vagina put me off. It was a big thing and I just can't imagine going through the process another time. Some of my friends opted for other methods but my husband convinced me that this was the best option for us".* **Couple 7-BKY-20230331**

***Women can't easily access the health facilities for IUDs***: Most of the women and their partners expressed concerns that some women in need of contraception are unable to access the healthcare facilities because of transportation challenges. Though some women desired to use the IUDs, the facilities offering the family planning methods were distant. Some of the women and their partners distant from the health facilities accept their fate and continue to have more children than they desire.

*"That is true, people come from far, we get a motorcycle at 8,000 Ugx coming and going back is also 8,000 Ugx. So, a couple thinks about that money and decides, no we shall not go! let us just give birth, God will know what to do for us. They end up giving birth nonstop because of that reason'.* **Couple 2-BKY-230331**

**Interpersonal factors.** ***Partner's refusal to use postabortion IUDs***: Most women and their partners reported that some men refused their partners to use IUDs for fears that their wives would become promiscuous as they had a means to stop them from conceiving yet they would be sexually active. Some young men in attendance expressed fear that the IUDs had the potential to cause 'an electric shock' to the penis during intercourse. There was also a belief that IUDs could prick them and cause them injuries during sex. These fears made some women and their partners hesitant to take on the IUDs.

*"I hear people say that when you are having sex, there is a time when you knock on the IUD and it produces some kind of electricity that will shock your penis. I have not experienced that yet. I'm worried of what I would do if it ever happens to me".* **Couple 3-BKY 17022023**

**Women's and their Partners' Experiences while using the Postabortion IUDs**

Majority of the women and their partners expressed differing views in regards to their experiences while using the their postabortion IUDs.

**"Postabortion IUDs improved the sexual life".** ***IUDs increase sexual lubrication***: Unlike other prior family planning methods like Depo-Provera (DMPA), majority of the women could relinquish their experience of having more lubrication with the postabortion IUDs. The pains during sex vanished. Some of the couples were happy to even recommend IUDs to their peers.

*"It is good. I have no challenges with it (IUD). The IUD doesn't make her dry yet the other family planning methods were making her dry like the injecta-plan. She didn't have sexual appetite. You would even feel that you don't like your wife any more".* **Couple 3-BKY-17022023**

Some of the women also articulated that with the birth control pills and the Depo-Provera injections, their sexual desires immensely reduced. This was further compounded by the reduction in their sexual lubrication as expressed in the excerpt below.

*"The birth control pills could make you lose sexual appetite but now I am okay. My experience with the Depo injection wasn't also that good. It used to make me lose sexual fluids but this* IUD *just increases the fluids".***Couple 1-MTN-20230331**

***Assurance of early return to fertility on removing IUDs***: Majority the women and their partners interviewed reported that they were reliably informed that fertility returned immediately in nearly 90% of the women and their partners after removing the IUDs. Having this assurance of immediate return to fertility after IUD removal, was such a relief especially that family planning methods were perceived to cause infertility in their communities.

*"I felt good because I thought that I will not have sex again with her or be able to impregnate her. The nurse told me that when the time comes and we want to give birth, she will remove the IUD. The economy now is not good and you have to prepare first before you can give birth again".* **Couple 2-MTN-20230330**

With the comprehensive family planning counselling offered by the study team, majority of the women embraced post-abortion IUDs. They were willing to delay their next pregnancies with no fears of the subsequent delays in return to fertility as expressed in the following excerpts.

*"The fact that I am going to get pregnant at the time that I want, and I can achieve that when I use the IUD. I liked the fact that you don't get unwanted pregnancies".* **Couple 7-BKY-20230331**

***Lesser side effects with IUDs compared to other contraceptive methods***: Compared to other family planning methods, most women mentioned they experienced lesser side effects while using postabortion IUDs. Some of women reported that there was lesser weight gain, menstrual irregularities while using the IUDs as compared to Depo-Provera injections. Such side effects caused tension and frustrations in their homes. Other women and their partners enunciated not even experiencing any side effects with the IUDs.

*"The other thing is that this IUD has had no side effects so far. It (IUD) is very fine. The menses are regular. Before when I was using another family planning method, we would be wanting to have sex and periods come. Now my husband is happy. The pills also could give me headaches. They made me add so much weight. All this is history now".* **Couple 1-BKY-20230331**

***Pelvic infections lessened with IUD use***: Some women also noticed that the occurrence of abnormal vaginal discharges reduced with the IUDs. This observation was more among women that opted to use the Levonorgestrel IUDs. The women and their partners expressed their gratitude to the research team for helping them fix the abnormal discharges. Some of the women reported that their confidence was restored as they were no longer concerned that an offensive smell was left behind as they went around doing their daily activities.

*"The hormonal IUD prevents genital infections. Us women, we have different sex partners, I will not hide that! So, if I go to another man, I come back to my husband when I am safe. I'm also no longer bothered that I have this foul-smelling discharge when I go to the market. I was so tired of going to hospital to treat those endless infections. The IUD is such a relief!"* **Couple 3- BKY 17022023**

**"Postabortion IUDs altered menses and caused sexual inconvenience".** Some of the women and their partners expressed frustrations with the postabortion IUDs. Such frustrations included the menstrual irregularities that followed the IUD insertions, making it hard for the women to plan on when to carry sanitary pads along. The IUDs were reported to prick husbands during sex and this caused tension in some families. Quite often the IUDs were reported to cause unbearable abdominal pain.

*Challenges in the menstrual cycles after IUD insertion*: Some women noted that they couldn't even judge when to have sex because the menstrual flow would just surface to the frustration of their husbands. This caused gratuitous tension in their families. This observation was more common among women using the copper IUDs.

*"No, I got challenges with my periods after the copper IUD was inserted. It can make you miss periods and when you skip one month, the next month, it comes with clots. This is so frustrating especially when your husband really wants to be intimate with you. You have to say 'No, not now! He usually gets so mad. I just can't tell how long this will go on".* **Couple 2-MTN-20230330**

*Pricks from IUD strings during sexual intercourse*: Some women disclosed that they had their IUDs inserted without their husbands' knowledge. This at times it turned out to be a nightmare when their husbands reported pricks during sex. Some husbands ended up recommending that their wives remove the IUDs. Some men ended up taking their wives back to health facilities to have more information about the IUDs. At times the IUD strings were shortened after comprehensive counselling. Other times, the IUDs were removed according to the discussants.

*"I feared because I inserted the IUD without telling him. When I returned home, he wanted to have sex. When we had sex, he was pricked. He asked me what it was, I told him to first finish then I explain. I told him and he feared and asked where I insert the IUD from. I told him the healthcare provider who did it. He called her and she confirmed that we did. I went with him and the nurse talked to him. The nurse shortened the strings and assured my husband that that experience won't reoccur. I was so scared but that was it".* **Couple 1-MTN-20230330**

*Unbearable abdominal pain following the IUD insertion*: Some women reported experiencing lower abdominal pains following the insertions of the IUDs. Some of the women related the pains to menstrual cramps. This abdominal pain at times made it hard for women to go around with their routine activities. The discussants associated this pain with the IUDs and were hoping that with time, their bodies would get used to them and enjoy the said benefits as emphasized by the study team.

*"At first, I was not doing my duties because of that abdominal pain that set in after the IUD was inserted. The feeling of the pain is like the cramps I experience during my menses. I'm hopeful that with time, I shall get used to it. I know it is the new norm now. I'm praying it doesn't take forever to go away that I get the feeling of what the nurse said while I'm using the method.* **Couple 3-KWP-20230330**

**Motivators to use of IUDs and women's and their partners' recommendations to improve the uptake of the Postabortion IUDs**

Most women expressed that having men involved in the process was pivotal in the uptake and use of the IUDs. The continued support from the healthcare providers through provision of holistic family planning counselling and IUD provision gave women and their partners the assurance that they were safe to use the postabortion IUDs. Women and their partners articulated that they felt fully in-charge of their reproductive lives to decide on when to have or delay childbirth. The desire to prevent repeated miscarriages in the subsequent pregnancies was another motivator to the uptake of the IUDs. Postabortion

IUDs were reported to have lesser side effects yet with early return to fertility after the IUD removal as compared to other contraceptive methods like Depo-Provera. Among the recommendations that women and their partners reported to improve the uptake of postabortion IUDs included; continued comprehensive family planning counselling and support from the health-care providers, community sensitization to demystify the myths and misconceptions that surround family planning methods especially the IUDs. Use of couple champions to motivate peers in the community to take up IUDs, and male involvement can go a long way in stirring the uptake of postabortion IUDs. The women and their partners also recommended women and their partners to desist from rumors if at all they were to take on the IUDs and ripe the allied benefits.

**"Motivators to the utilization of Postabortion IUDs".** *Male involvement in postabortion IUD counselling and provision*: Some of the participants mentioned that when men were brought onboard in the regards to the uptake and use of the IUDs, their mood and support improved. When women and their partners are given family planning counselling together, the uptake was reported to be better than when women took on the mandate in their own hands. Nonetheless, most men acknowledged that it was the women that carry the burden of the pregnancies and had the powers to decline when not carry the pregnancies. They also believed that if men didn't agree that the women should take on the IUDs, nothing would move forward.

*"It is okay because women have autonomy over their bodies. When they decide to give birth, they do. If they don't want, it is still okay. We always make decisions together. We discuss and always reach an agreement. To me, when you are in agreement with one another, nothing can fail because you have to decide together".* **Couple 2-MTN-20230330**

*Recommendation of IUDs by healthcare providers as the best contraceptive option*: Most of the women and their partners emphasized that the family planning counselling they received from the healthcare providers was pivotal in their decision to take up the postabortion IUDs. They perceived the healthcare providers' advice to hold so much as they were knowledgeable about the available contraceptive choices for the women and their partners. The bond and trust they had in their healthcare providers couldn't be underestimated. The relationship they held with healthcare providers was the assurance they had then that even when any complications arose from the IUDs, they would handle the women thus their motivation to choosing the IUDs over the other methods.

*"The healthcare provider educated us on contraception. We were many but I picked an interest. I went and informed my husband. This is how we chose the IUD over the other methods. For this reason, no other person can influence my decision because they are just people like me. They are not experienced in IUDs. If it is the healthcare provider telling me to use this IUD because of the advantages, I can follow her advice".* **Couple 1-KGD-20230331**

*Women and their partners being in charge of their reproductive life*: The ability to hold the driving seat in determining when to and not to have children, motivated women and their partners to take on postabortion IUDs. The IUDs were deemed to be more effective yet with lesser side effects as compared to the short-acting family planning methods. This control over the reproductive life, motivated the participants to take up IUDs and even recommend them to their peers.

*"The assurance that I am going to get pregnant at the time that I want and I can achieve that when I use the IUD, motivates me to use it (IUD). I liked the fact that you don't get unwanted pregnancies. I also know that there are lesser side effects like low libido, weight gain and headaches with the IUD as compared to Depo injections".* **Couple 7-BKY-20230331**

**Women's and their partners' recommendations to improve the uptake of the postabortion IUDs**

**Continued comprehensive counselling and support from healthcare providers.** Healthcare providers play a key role in the uptake of the postabortion IUDs according to the women and their partners. Their knowledge is vital in

dispelling the myths and misconception around the IUDs. Maintaining a strong bond with women and their partners using the postabortion IUDs can bridge the gap between the healthcare providers and the community. The assurance that in case of any complications, the healthcare providers are readily available for such women and their partners, has the potential to scale up the uptake of IUDs in the community.

*"I think the healthcare providers are trained and they give you something they know can work for you. The same with me, I know there is always someone who is an expert in certain things. When it comes to health, the healthcare providers are very vital in this. You cannot go to the neighbour yet the health care provider is there. I cannot ask my neighbour because she is not educated. So for any challenge with the IUD, I just go to the healthcare provider for advice. She has always been there for us".* **Couple-01-KWP-17022023**

***Use of women and their partners using the Postabortion IUDs as champions***: According to some discussants, having women and their partners using the postabortion IUDs to reach out to peers in the community, would improve the uptake of the IUDs. Having experienced the benefits of the IUDs, it would be easier for the peers to learn from them and opt to give the postabortion IUDs a chance to prevent unwanted and unintended pregnancies. The women and their partners advocated having talks in the market places, over radios to share their lived experiences while using the postabortion IUDs. In so doing, the women and their partners emphasized that this approach would help to demystify misconceptions that spread like wild fire that IUDs cause infertility, cancers and deaths.

*"Yes, if we find those who need it, we can educate them to use it. We are happy and we are doing our work to recommend* IUDs *to others. I also tell other men to use family planning. I advised one couple to use family planning. They went and also got it, it is now two months and they have no challenges with it".* **Couple 1-MTN-20230331**

*"They also told us to recommend it to the other women in the villages. I got two women. I educated them about* IUDs. *I brought them to the healthcare provider. They inserted it and they told me they have no challenges so far with it. The women need to be sensitized. There is a need for outreaches because women don't come to the health facility. They only come when they are sick".* **Couple 2-KGD-20230330**

***Community sensitization on the use of Postabortion IUDs***: Most of the participants recommended that community outreaches should be utilized to reach out to the communities with evidence-based literature to all stakeholders. In so doing, the communities will be educated on the benefits of child spacing and the utilization of the IUDs. The discussants also emphasized the relevance of teaching the communities where to find the different family planning methods and how the women and their partners can be helped when faced with complications resulting from the different family planning options. They also advised use of social media platforms like WhatsApp, TikTok, and Facebook to reach out to more women and their partners on the benefits of postabortion IUDs.

*"Another thing is to sensitize the community on IUDs. If a healthcare provider can come to the community and mobilize people and sensitize them. After that, the people can ask questions privately about this method. You do it over the radio, TikTok, or Facebook. What most people have is WhatsApp to keep teaching couples on the relevance of using these IUDs to stop having unwanted pregnancies".* **Couple 1-KWP-20230331**

***Ignoring rumors on IUDs***: From their experiences, most of the participants advised that women and their partners in the communities, to avoid rumors about family planning methods and always embark on using the proper channels to get the correct information. According to them, most of the information they had gathered from their peers was wrong. If the women and their partners never seek a second opinion, they will continue to have more children.

*"What I understand most is not to listen to people's words. Most times when you listen to what people say, you can get in trouble. You have to only listen to healthcare providers. If you are discussing in your home, no one can influence you. We started using an IUD after being health educated about it. The nurse calls us often to check on us. Our advice is for couples to ignore rumors and consult healthcare providers to avoid getting unwanted pregnancies".* **Couple 1-BKY-17022023**

## Discussion

This study set out to explore the perceptions of women and their partners on the utilization of postabortion intrauterine contraception after medical management of first trimester incomplete abortion. From the content analysis as noted in the Andersen's Behavioral Model of Health Services [45], the key predisposing factors included prior contraceptive experiences, and the communication within the marriage. The barriers to utilization of IUDs included myths and misconceptions on IUDs at community and individual level, spouse refusal of IUDs, IUD side effects, religious and cultural misbeliefs on contraception. The economic status of the women and their partners at times led to transport challenges that made utilization of IUDs impossible. Most of the discussants emphasized that spousal approval of the method, peer influence, free access to IUDs, freedom from prior contraceptive side effects, approval of the method by healthcare providers, increased sexual lubrication and early return to fertility after IUD removal, were the key enabling factors for the uptake of the IUDs. The perceived need factors reported by most women and their partners were; the desire to avoid adverse pregnancy outcomes and also delay the subsequent pregnancy without any fears of future infertility.

Our study findings on spousal refusal, myths and misconceptions as barriers to the utilization of the IUDs, are consistent with what has been reported in Imo State in Nigeria [57], where the uptake of the IUDs was 2.5% among the participants. As noted by other researchers [26,58,59], the fear of side effects such as menstrual irregularities [60], weight changes [23], loss of sexual drive that come along with the IUDs, keep women and their partners from using the method. The fear of side effects also drives women and their partners to use traditional methods as they are perceived to be safer options, thereby limiting the uptake of IUDs [18]. Spousal refusal as noted in our study, is key in stopping many women and their partners from utilizing the IUDs even when provided at no costs to women and their partners.

Communication in homes is key and whenever there's a breakdown in the exchange of the information on the reproductive priorities as perceived by the men, the utilization of IUDs declines [61,62]. From the study findings, some of the women had the IUDs inserted without their husbands' knowledge but on discovery, there was undue tension in the families prompting IUD removal. Men expressed their views that once they were brought on board, they were more supportive on the utilization of the IUDs. This observation was reported by Ogunjuyigbe in Southwest Nigeria [62]. The fear of infertility was fronted by women and their partners in our study as a major barrier to the utilization of postabortion IUDs. Other researchers have reported a similar trend on misconceptions and myths on IUDs among their respondents as a major drawback to the uptake of IUDs [21–23]. This calls for a behavioral change packaging of the information that is offered to the women and their partners to demystify these fears if at all the uptake of IUDs is to improve from two percent as reported in Uganda [14].

Though women and their partners had differing opinions on the impact of religion on the uptake of the IUDs, some of the women and their partners in central Uganda believe they hold the mandate of multiplying to fill the earth as commanded by God. The religious values still pose a threat to the utilization of postabortion IUDs. Similar findings about the impact of religion on uptake of contraception have been reported by other researchers [20,57] to be a hindrance to the uptake of IUDs. Despite their religious beliefs, some women and their partners in our study, as reported by Maqbool among postabortion clients in Sargodha, Pakistan [63] opted to use postabortion IUDs due to the economic pressure and desire to delay childbirth.

Though cultural norms and beliefs didn't have impact on the uptake of the IUDs in our study, socio-cultural beliefs and norms are reported to be pivotal in the utilization of family planning methods [20,28,29]. Socio-cultural beliefs tended to

cause a power imbalance exhorting men over women which ultimately made men adamant to take on family planning methods [18]. Our findings could have differed because the study was conducted only in central Uganda with tribes mainly from the Bantu origin. Inclusion of other regions would have given a broader perspective on the influence of socio-cultural norms on the uptake of IUDs as there would have been more tribal engagements.

In our study, increased sexual lubrication was reported as one of the positive experiences. According to Olge [64], hormonal IUDs, were associated with lesser sexual pains as compared to other contraceptive methods. There was however, no association of the hormonal IUDs and other aspects of sexual function such as lubrication, arousal or orgasm. Other studies [65,66] have however reaffirmed our findings of increased sexual lubrication noting improvement in the overall sexual function among users of the Levonorgestrel IUDs. Levonorgestrel binds to the androgen receptor leading to production of glycoproteins required for the mucus production and increased lubrication [67]. The increase in the vaginal and cervical mucus is also hypothesized to increase the genital arousal that ultimately turns on the arousal centrally improving the quality of sexual life for the women and their partners [68]. Our finding could also follow the assurance that with the postabortion IUDs, the fear of unwanted pregnancies subsided and this improved the quality of the sexual life thereby increasing the observed lubrication [69,70]. Further data is however needed as there is conflicting information on the impact of IUD use on sexual lubrication [64,71–73].

Postabortion IUDs were also reported to have lesser side effects such as weight gain, and menstrual irregularities as compared to other contraceptives. According to Ortayli [74], after immediate postabortion contraception use, there was significant weight gain associated with implants as compared to the IUDs. Weight gain also associated with Depo-Provera [75] has been one of the causes of high discontinuation rates of hormonal contraception. Postabortion IUDs have been reported to have barely any associated weight gain as compared to implants or other short-acting hormonal contraceptives [76,77], hence explaining the perceived positive experiences among the women and their partners in our study.

With prior experience of heavy periods associated with Depo-Provera, once the women and their partners experienced lighter, less painful periods and occasionally no menses especially with the Levonorgestrel IUDs [78,79], they perceived it as positive experience and were willing to continue with the method. Menstrual changes especially heavy or intermenstrual bleeding have been associated with high dissatisfaction and discontinuation rates of contraception [80]. These menstrual changes are also reported to cause concerns on the effectiveness, sexual life, and future fertility among users [81]. Understanding the women's and their partners' needs and socio-cultural context is therefore paramount in improving the uptake of postabortion IUDs, to build the trust in the method. This calls for policy makers to use this knowledge to design policies and programs that are customized to the women's and their partners' needs and socio-cultural contexts of the target communities if at all successful uptake of IUDs is to be observed.

Though IUDs were reported before to increase the chances of pelvic infections especially in the first few months following their insertion [81], participants in our study reported otherwise. Women in our study reported that with the postabortion IUDs, pelvic infections reduced. On the contrary to our findings, IUDs have been reported before to increase the risk of pelvic infections especially in the first three weeks following their insertion. Postabortion IUDs especially the Levonorgestrel IUDs gave the women and their partners in the study a positive experience of reduced pelvic infections. Levonorgestrel cause thickening of cervical mucus which ultimately reduces the frequency of ascending pelvic infections [82].

Among the negative experiences reported in our study, included menstrual irregularities that made it hard for women and their partners to plan on when to have sexual intercourse especially among women who did not want to have sex during menses. This caused frustrations among some women and their partners raising tension in homes. IUDs were reported to cause menstrual irregularities such as intermenstrual bleeding, heavy and painful periods. These concerns were common among the Copper T as compared to the Levonorgestrel IUD users as reported by other researchers leading to 15% of the users removing the IUDs [83]. So as to overcome this negative experience, healthcare providers need to offer comprehensive family planning counselling to the potential users of the likely side effects but with the assurance that the menstrual changes might improve with time especially after the six months [84].

Men reported to be pricked by the IUD strings and this didn't go well especially among women and their partners when women inserted the IUDs without the consent of their husbands. Some men expressed a fear that the IUDs could cause 'electric shock' that would affect their manhood in the long run. Similar findings were reported in Zimbabwe where pricks from IUD strings were reported to discourage men from using them in their families possibly contributing to the 0.8% IUD user rate [85,86]. Women have devised means of proceeding with the IUD insertions without the knowledge of their spouses to achieve their reproductive goals [18,85]. Healthcare providers should ensure that the IUD strings are cut not too short but at 2 cm.

The assurance of early return to fertility was viewed as a key motivator for women and their partners opting for postabortion IUDs over injectable Depo-Provera and sterilization. With concerns about subsequent fertility, it's prudent that women and their partners are given the right information on the effect of contraceptive use on their fertility [87,88]. There have been concerns that the Copper T IUDs could lead to secondary infertility through the associated pelvic infections [89]. Levonorgestrel IUDs have also been implicated to cause temporary infertility through suppression of the hypothalamo-pituitary-ovarian axis to cause anovulation [90]. Current evidence disapproves these arguments suggesting early return to fertility [91]. Mansour reported that the pregnancy rates after termination of IUDs, was been 86.1–92.3% which is comparable to the natural method users and non-users [92]. Healthcare providers can therefore use this evidence to encourage women and their partners to use postabortion IUDs without much concerns that their fertility might be delayed because of the method.

Male involvement was reported to be a pertinent motivator in the uptake of the postabortion IUDs [93,94]. Measures to improve IUD uptake ought to understand the social context women operate in. Therefore, policies that target male social networks can be explored to reach men with the right information in regards to the benefits of postabortion IUDs [18]. Maintaining healthy relationships with the healthcare providers, was another motivator to improving the uptake of postabortion IUDs. Healthcare providers play a pivotal role in the uptake of postabortion IUDs, their competences, age and messages as they talk about contraception can either stir or block women and their partners from using IUDs [95]. Healthcare providers' gaps in knowledge and skills can fuel the misconceptions and stop women and their partners from choosing the IUDs [16,96–98]. This calls for policy makers to ensure that the healthcare providers are skilled and knowledgeable in all contraceptives available.

To improve the uptake of postabortion IUDs, use of champion couples to encourage others to take on the methods, could be explored. Other community outreach measures such as use of social media platforms TikTok, WhatsApp, Facebook, and radio talks, can be explored to reach the communities with evidence-based information as recommended by other researchers [18,26,29,93].

## Strengths

Our study was conducted in a setting where all materials and costs involved in the access of the postabortion IUDs, were covered by the study team. In so doing some of the economic barriers were minimized. The interviews involved women and their partners as opposed to individual spouses, giving a rich diversity of knowledge that might have given a richer reflection on the socio-dynamic factors that influence the perceptions on postabortion IUDs. We ensured that there was room for free expression even when the discussants contested each other's views with the expertise of the research team in qualitative enquiry. It's however not clear whether the joint interviews might not have affected expression of the individuals in regards to the power dynamics in the society. We used in-depth interviews as opposed to focus group discussions and this allowed free expression of women and their partners without fear of what others women and their partners commented on the individual family dynamics in regards to contraception. Field notes were taken for triangulation.

## Limitations

The study was undertaken in central Uganda, and might not be a true reflection of the women and their partners from other regions of the country. The tribes in central Uganda are mainly of the Bantu origin. Inclusion of other regions with

different tribal origins would have given a broader perspective on the influence of socio-cultural norms on the uptake of IUDs. We however conduct the study as rigorously as possible to ensure generalizability of the findings.

Our study could have been more informative if we also explored the perceptions of women and their partners that declined to take on the postabortion IUDs despite the availability of all the materials. Interviewing men and women separately in regards to uptake of postabortion IUDs, would have improved our understanding of the socio-cultural dynamics that influence contraceptive choices but due to unavoidable challenges, their opinions were not taken into account. We interviewed only women and their partners who were using the postabortion IUDs after medical management of incomplete abortions. Interviewing women and their partners' who had expelled their IUDs or had discontinued IUDs use, would have been informative on other determinants of the uptake and utilization of postabortion IUDs. We also acknowledge that reporting findings as the general perceptions per the couples deprived us of the individual divergent perspectives of the women or their partners on some of the pertinent issues on the utilization of postabortion intrauterine contraception.

## Conclusion

Understanding the socio-cultural dynamics of women and their partners, is pivotal in the uptake of postabortion IUDs. Perceived barriers like myths and misconceptions, fear of IUD side effects, spouse refusal, transport restrictions, need to be addressed to improve use of postabortion IUDs by women and their partners. Healthcare providers should provide evidence-based family planning counselling and provision for women and their partners, to demystify individual and community misconceptions on IUD use. Male involvement, the assurance of early return to fertility after IUD discontinuation, use of champion couples utilizing postabortion IUDs, and community sensitization outreaches on postabortion IUDs, are nuggets that can be explored to enhance the uptake of IUDs. Policy makers need to encourage pre- and in-service trainings that ensure that the healthcare providers are skilled and knowledgeable in all contraceptives available, to demystify misconceptions on intrauterine contraception.

## Appendix 1

**Descriptive characteristics of five selected health facilities in central Uganda**

| Health Unit | Location | Bed capac-ity | No. of healthcare providers | Duration of work | Cumulative 1st tri-mester Abortion load in the past 3 months | Setting | Service delivered |
|---|---|---|---|---|---|---|---|
| Kawempe National Referral Hospital | Kampala | 900 | 500 | 24 hours a day, 7 days a week. | 484 | Urban | Teaching hospital. Offers free emergency gynaeco-logical and obstetric services. |
| Mityana General Hospital | Mityana | 100 | 116 | 24 hours a day, 7 days a week. | 100 | Peri-urban | Offers comprehensive maternal health services including PAC. |
| Kiganda Health Centre IV | Kas-sanda | 15 | 50 | 24 hours a day, 7 days a week | 38 | Rural | Offers both in and outpatient ser-vices including family planning. |
| Kayunga Regional Referral Hospital | Kayunga | 200 | 179 | 24 hours a day, 7 days a week | 212 | Peri-urban | Offers specialist maternal ser-vices including PAC. |
| Bukuya Health centre III | Kas-sanda | 20 | 30 | 24 hours a day, 7 days a week | 60 | Rural | Offers maternal services including PAC, family planning services. |

## Acknowledgments

The lead author along with the co-authors extend their gratitude to the participants, doctoral committee and the research team that enable this study to become a reality. This study was undertaken as part of the lead author's doctoral dissertation.

## Author contributions

**Conceptualization:** Herbert Kayiga.

**Data curation:** Herbert Kayiga.

**Formal analysis:** Herbert Kayiga.

**Funding acquisition:** Kristina Gemzell-Danielsson.

**Methodology:** Herbert Kayiga, Emelie Looft-Trägårdh.

**Project administration:** Herbert Kayiga.

**Software:** Herbert Kayiga.

**Supervision:** Othman Kakaire, Nazarius Mbona Tumwesigye, Musa Sekikubo, Joseph Rujumba, Josaphat Byamugisha, Kristina Gemzell-Danielsson.

**Validation:** Herbert Kayiga.

**Writing – original draft:** Herbert Kayiga.

**Writing – review & editing:** Herbert Kayiga, Emelie Looft-Trägårdh, Othman Kakaire, Nazarius Mbona Tumwesigye, Musa Sekikubo, Joseph Rujumba, Josaphat Byamugisha, Kristina Gemzell-Danielsson.

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
