## [Decision Letter · Decision Letter 0]

Dear Dr. Kayiga,

Thank you for submitting your manuscript to PLOS ONE. After careful consideration, we feel that it has merit but does not fully meet PLOS ONE’s publication criteria as it currently stands. Therefore, we invite you to submit a revised version of the manuscript that addresses the points raised during the review process.

We look forward to receiving your revised manuscript.

Kind regards,

Adnan Ahmad Khan, MBBS, MS.

Academic Editor

PLOS ONE

**Journal Requirements:**

1. When submitting your revision, we need you to address these additional requirements. Please ensure that your manuscript meets PLOS ONE's style requirements, including those for file naming. The PLOS ONE style templates can be found at https://journals.plos.org/plosone/s/file?id=wjVg/PLOSOne_formatting_sample_main_body.pdf and https://journals.plos.org/plosone/s/file?id=ba62/PLOSOne_formatting_sample_title_authors_affiliations.pdf 2. Thank you for stating in your Funding Statement: This project was supported by funds from the Swedish Research Council, (Grant 2019-04256) in partnership with Makerere University and the MakRif Project. The content is solely the responsibility of the authors and does not necessarily represent the official views of The Swedish Research Council, Makerere University or the MakRif Project. The funders provided support in the form of research expenses, but did not have any additional role in the study design, data collection and analysis, decision to publish, or preparation of the manuscript.  Please provide an amended statement that declares *all* the funding or sources of support (whether external or internal to your organization) received during this study, as detailed online in our guide for authors at http://journals.plos.org/plosone/s/submit-now.  Please also include the statement “There was no additional external funding received for this study.” in your updated Funding Statement. Please include your amended Funding Statement within your cover letter. We will change the online submission form on your behalf. 3. Thank you for stating the following in the Acknowledgments Section of your manuscript: The lead author along with the co-authors extend their gratitude to the participants, doctoral committee and the research team that enable this study to become a reality. This study was undertaken as part of the lead author’s doctoral dissertation funded by the collaboration between The Swedish Research Council and Makerere University in partnership with MakRif project. We note that you have provided funding information that is not currently declared in your Funding Statement. However, funding information should not appear in the Acknowledgments section or other areas of your manuscript. We will only publish funding information present in the Funding Statement section of the online submission form. Please remove any funding-related text from the manuscript and let us know how you would like to update your Funding Statement. Currently, your Funding Statement reads as follows: This project was supported by funds from the Swedish Research Council, (Grant 2019-04256) in partnership with Makerere University and the MakRif Project. The content is solely the responsibility of the authors and does not necessarily represent the official views of The Swedish Research Council, Makerere University or the MakRif Project. The funders provided support in the form of research expenses, but did not have any additional role in the study design, data collection and analysis, decision to publish, or preparation of the manuscript.  Please include your amended statements within your cover letter; we will change the online submission form on your behalf. 4. In the online submission form, you indicated that “The dataset used and analyzed in this study are also available with the corresponding author on request.” All PLOS journals now require all data underlying the findings described in their manuscript to be freely available to other researchers, either a. In a public repository, b. Within the manuscript itself, or c. Uploaded as supplementary information.This policy applies to all data except where public deposition would breach compliance with the protocol approved by your research ethics board. If your data cannot be made publicly available for ethical or legal reasons (e.g., public availability would compromise patient privacy), please explain your reasons on resubmission and your exemption request will be escalated for approval.

**Additional Editor Comments:**

Thank you for submitting this study which addresses an important knowledge gap. Overall the study is well conducted and analyzed. There are the following minor revisions that may be considered.

1. While the overall manuscript is well written there are typos and some logical flow errors. Example dare instead of dire on line 54, additional word "yet" on line 70 etc. Please run word editor to ensure better readability

2. Interviews were conducted with only English speakers. This may have introduced bias towards higher educated or better socio-economic couples. Please address this in limitations including how this may impact the interpretation of your results

3. Since both husband and wife were interviewed, it would be interesting to see how their perceptions were similar and different and on what issues

**Reviewer 1:**

I commend the authors’ efforts in trying to fill an important gap in research and share such important findings. After reviewing the manuscript, here are some important points that the authors should consider:

• This sentence needs revision as unintended pregnancies are not necessarily unwanted pregnancies. They are unplanned. This means the pregnancies were not intended at the time of conception, regardless of whether they were unwanted or mistimed. This argument has evolved over time. Use updated articles.

• The argument on not well-documented reasons should be reviewed - “Despite proven effectiveness and safety of intrauterine contraception yet with a failure rate of less than one percent [14], uptake among Ugandan women has stagnated below two percent among married and 15% among sexually active unmarried women [15] for reasons that may not be well documented”. A quick Google search reflects otherwise (https://bmcwomenshealth.biomedcentral.com/articles/10.1186/s12905-022-01856-1;
https://pmc.ncbi.nlm.nih.gov/articles/PMC10022157/)

• The paragraphs after lines 72 and 73 are confusing. The previous argument was that there may not be well documented reasons for the stagnation in the uptake of intrauterine conception among Ugandan women. However, subsequent arguments in the following paragraphs stated otherwise. I think these paragraphs should be revised to reflect exactly what the authors are trying to present in the introduction.

• I noticed that many of the references cited in the manuscript are somewhat dated. While these foundational works are undoubtedly valuable, it is also essential to incorporate more recent research to provide a comprehensive and current perspective on the topic. I recommend updating the references with more recent studies that have built upon or expanded the concepts discussed in the older articles.

• “Using purposive sampling with maximal variation, fifteen women and their partners aged between 18 and 49 years were invited to participate in the study at five public facilities in central Uganda.” What does maximal variation mean? I appreciate the descriptive details of the facilities, but I do not see the relevance in the study. This may be useful for other reports on the larger RCT study. Could you share the inclusion criteria to further highlight the level of variation?

• It was stated that the sample size was influenced by assumed saturation. Could this be clarified? Could the authors elaborate on what influenced the sample size?

• Despite the subtitle, participant recruitment details were not adequately presented in the manuscript. How were participants recruited? For the few details that were included, there was no reference to whether participants had previously consented to being contacted before contact details were retrieved from the database.

• “Fifteen in-depth interviews were conducted in English or Luganda by two interviewers and two note takers with prior experience in qualitative inquiry”. I appreciate the use of either English or Luganda to enhance quality data. Could you share why these options were presented? How did the subsequent translation of documents from one language to another impact data and its analysis? How did you mitigate the loss of the meaning of ideas during translation?

• I’m curious to know the influence of the socio-demographic details such as age and education on participants’ decisions.

• I appreciate that the analysis and the codebook development were adequately presented - showing how the codebook (subthemes and themes) was developed gives credence to this qualitative work.

• The verbatim quotes under cultural beliefs on IUDs do not necessarily support the arguments in that theme. Some of the quotes are more like responses to barriers that were presented to them, rather than the barriers that were shared by the participants.

• Review statement – “We used a semi-structured interview guide to collect information on participants’ sociodemographic characteristics, and the various barriers and motivators in the utilization of the IUDs”. I suggest overall proofreading and manuscript editing.

• A more detailed conclusion should be considered.

• Overall, the authors have presented important findings that will contribute to ongoing arguments on postabortion intrauterine contraception in Uganda and the global context if the manuscript is reviewed adequately before publication.

Reviewers' comments:

Reviewer's Responses to Questions

**Comments to the Author**

1. Is the manuscript technically sound, and do the data support the conclusions?

Reviewer #1: Yes

Reviewer #2: Yes

2. Has the statistical analysis been performed appropriately and rigorously?

Reviewer #1: N/A

Reviewer #2: Yes

3. Have the authors made all data underlying the findings in their manuscript fully available?

Reviewer #1: Yes

Reviewer #2: Yes

4. Is the manuscript presented in an intelligible fashion and written in standard English?

Reviewer #1: Yes

Reviewer #2: Yes

**Reviewer #1:**  I commend the authors’ efforts in trying to fill an important gap in research and share such important findings. After reviewing the manuscript, here are some important points that the authors should consider:

• This sentence needs revision as unintended pregnancies are not necessarily unwanted pregnancies. They are unplanned. This means the pregnancies were not intended at the time of conception, regardless of whether they were unwanted or mistimed. This argument has evolved over time. Use updated articles.

• The argument on not well-documented reasons should be reviewed - “Despite proven effectiveness and safety of intrauterine contraception yet with a failure rate of less than one percent [14], uptake among Ugandan women has stagnated below two percent among married and 15% among sexually active unmarried women [15] for reasons that may not be well documented”. A quick Google search reflects otherwise (https://bmcwomenshealth.biomedcentral.com/articles/10.1186/s12905-022-01856-1;
https://pmc.ncbi.nlm.nih.gov/articles/PMC10022157/)

• The paragraphs after lines 72 and 73 are confusing. The previous argument was that there may not be well documented reasons for the stagnation in the uptake of intrauterine conception among Ugandan women. However, subsequent arguments in the following paragraphs stated otherwise. I think these paragraphs should be revised to reflect exactly what the authors are trying to present in the introduction.

• I noticed that many of the references cited in the manuscript are somewhat dated. While these foundational works are undoubtedly valuable, it is also essential to incorporate more recent research to provide a comprehensive and current perspective on the topic. I recommend updating the references with more recent studies that have built upon or expanded the concepts discussed in the older articles.

• “Using purposive sampling with maximal variation, fifteen women and their partners aged between 18 and 49 years were invited to participate in the study at five public facilities in central Uganda.” What does maximal variation mean? I appreciate the descriptive details of the facilities, but I do not see the relevance in the study. This may be useful for other reports on the larger RCT study. Could you share the inclusion criteria to further highlight the level of variation?

• It was stated that the sample size was influenced by assumed saturation. Could this be clarified? Could the authors elaborate on what influenced the sample size?

• Despite the subtitle, participant recruitment details were not adequately presented in the manuscript. How were participants recruited? For the few details that were included, there was no reference to whether participants had previously consented to being contacted before contact details were retrieved from the database.

• “Fifteen in-depth interviews were conducted in English or Luganda by two interviewers and two note takers with prior experience in qualitative inquiry”. I appreciate the use of either English or Luganda to enhance quality data. Could you share why these options were presented? How did the subsequent translation of documents from one language to another impact data and its analysis? How did you mitigate the loss of the meaning of ideas during translation?

• I’m curious to know the influence of the socio-demographic details such as age and education on participants’ decisions.

• I appreciate that the analysis and the codebook development were adequately presented - showing how the codebook (subthemes and themes) was developed gives credence to this qualitative work.

• The verbatim quotes under cultural beliefs on IUDs do not necessarily support the arguments in that theme. Some of the quotes are more like responses to barriers that were presented to them, rather than the barriers that were shared by the participants.

• Review statement – “We used a semi-structured interview guide to collect information on participants’ sociodemographic characteristics, and the various barriers and motivators in the utilization of the IUDs”. I suggest overall proofreading and manuscript editing.

• A more detailed conclusion should be considered.

• Overall, the authors have presented important findings that will contribute to ongoing arguments on postabortion intrauterine contraception in Uganda and the global context if the manuscript is reviewed adequately before publication.

**Reviewer #2: ** The manuscript is well written and the research is conducted accordingly with methodological rigor.

There is no research or evidence of publication ethical issues in the this manuscript to the best of my understanding.

**Do you want your identity to be public for this peer review?** For information about this choice, including consent withdrawal, please see our Privacy Policy

Reviewer #1: No

Reviewer #2: No

---

## [Author Response · Author response to Decision Letter 1]

21 Apr 2025

COLLEGE OF HEALTH SCIENCES

SCHOOL OF MEDICINE

DEPARTMENT OF OBSTETRICS AND GYNAECOLOGY

16th April 2025

RESPONSE TO REVIEWS COMMENTS:

Object: PONE-D-24-58782 “Perceptions of Women and their Partners on Postabortion Intrauterine Contraception: A Qualitative study in central Uganda”

With great pleasure, I’m thankful for your comments towards our manuscript. In response to the reviewers’ comments sent to us on 11th April 2025, we have revised the manuscript accordingly.

Comment Response to Comment Page No. and Line

Journal Requirements:

This project was supported by funds from the Swedish Research Council, (Grant 2019-04256) in partnership with Makerere University and the MakRif Project. The content is solely the responsibility of the authors and does not necessarily represent the official views of The Swedish Research Council, Makerere University or the MakRif Project. The funders provided support in the form of research expenses, but did not have any additional role in the study design, data collection and analysis, decision to publish, or preparation of the manuscript.

The lead author along with the co-authors extend their gratitude to the participants, doctoral committee and the research team that enable this study to become a reality. This study was undertaken as part of the lead author’s doctoral dissertation funded by the collaboration between The Swedish Research Council and Makerere University in partnership with MakRif project.

This project was supported by funds from the Swedish Research Council, (Grant 2019-04256) in partnership with Makerere University and the MakRif Project. The content is solely the responsibility of the authors and does not necessarily represent the official views of The Swedish Research Council, Makerere University or the MakRif Project. The funders provided support in the form of research expenses, but did not have any additional role in the study design, data collection and analysis, decision to publish, or preparation of the manuscript.

4. In the online submission form, you indicated that “The dataset used and analyzed in this study are also available with the corresponding author on request.”

Additional Editor Comments:

Thank you for submitting this study which addresses an important knowledge gap. Overall the study is well conducted and analyzed. There are the following minor revisions that may be considered.

1. While the overall manuscript is well written there are typos and some logical flow errors. Example dare instead of dire on line 54, additional word "yet" on line 70 etc. Please run word editor to ensure better readability

2. Interviews were conducted with only English speakers. This may have introduced bias towards higher educated or better socio-economic couples. Please address this in limitations including how this may impact the interpretation of your results

3. Since both husband and wife were interviewed, it would be interesting to see how their perceptions were similar and different and on what issues

Reviewer 1:

I commend the authors’ efforts in trying to fill an important gap in research and share such important findings. After reviewing the manuscript, here are some important points that the authors should consider:

• This sentence needs revision as unintended pregnancies are not necessarily unwanted pregnancies. They are unplanned. This means the pregnancies were not intended at the time of conception, regardless of whether they were unwanted or mistimed. This argument has evolved over time. Use updated articles.

• The argument on not well-documented reasons should be reviewed - “Despite proven effectiveness and safety of intrauterine contraception yet with a failure rate of less than one percent [14], uptake among Ugandan women has stagnated below two percent among married and 15% among sexually active unmarried women [15] for reasons that may not be well documented”. A quick Google search reflects otherwise (https://bmcwomenshealth.biomedcentral.com/articles/10.1186/s12905-022-01856-1;
https://pmc.ncbi.nlm.nih.gov/articles/PMC10022157/)

• The paragraphs after lines 72 and 73 are confusing. The previous argument was that there may not be well documented reasons for the stagnation in the uptake of intrauterine conception among Ugandan women. However, subsequent arguments in the following paragraphs stated otherwise. I think these paragraphs should be revised to reflect exactly what the authors are trying to present in the introduction.

• I noticed that many of the references cited in the manuscript are somewhat dated. While these foundational works are undoubtedly valuable, it is also essential to incorporate more recent research to provide a comprehensive and current perspective on the topic. I recommend updating the references with more recent studies that have built upon or expanded the concepts discussed in the older articles.

• “Using purposive sampling with maximal variation, fifteen women and their partners aged between 18 and 49 years were invited to participate in the study at five public facilities in central Uganda.” What does maximal variation mean? I appreciate the descriptive details of the facilities, but I do not see the relevance in the study. This may be useful for other reports on the larger RCT study. Could you share the inclusion criteria to further highlight the level of variation?

• It was stated that the sample size was influenced by assumed saturation. Could this be clarified? Could the authors elaborate on what influenced the sample size?

• Despite the subtitle, participant recruitment details were not adequately presented in the manuscript. How were participants recruited? For the few details that were included, there was no reference to whether participants had previously consented to being contacted before contact details were retrieved from the database.

• “Fifteen in-depth interviews were conducted in English or Luganda by two interviewers and two note takers with prior experience in qualitative inquiry”. I appreciate the use of either English or Luganda to enhance quality data. Could you share why these options were presented? How did the subsequent translation of documents from one language to another impact data and its analysis? How did you mitigate the loss of the meaning of ideas during translation?

• I’m curious to know the influence of the socio-demographic details such as age and education on participants’ decisions.

• I appreciate that the analysis and the codebook development were adequately presented - showing how the codebook (subthemes and themes) was developed gives credence to this qualitative work.

• The verbatim quotes under cultural beliefs on IUDs do not necessarily support the arguments in that theme. Some of the quotes are more like responses to barriers that were presented to them, rather than the barriers that were shared by the participants.

• Review statement – “We used a semi-structured interview guide to collect information on participants’ sociodemographic characteristics, and the various barriers and motivators in the utilization of the IUDs”. I suggest overall proofreading and manuscript editing.

• A more detailed conclusion should be considered.

• Overall, the authors have presented important findings that will contribute to ongoing arguments on postabortion intrauterine contraception in Uganda and the global context if the manuscript is reviewed adequately before publication.

Reviewers' comments:

Reviewer's Responses to Questions

Comments to the Author

1. Is the manuscript technically sound, and do the data support the conclusions?

Reviewer #1: Yes

Reviewer #2: Yes

2. Has the statistical analysis been performed appropriately and rigorously?

Reviewer #1: N/A

Reviewer #2: Yes

3. Have the authors made all data underlying the findings in their manuscript fully available?

Reviewer #1: Yes

Reviewer #2: Yes

4. Is the manuscript presented in an intelligible fashion and written in standard English?

Reviewer #1: Yes

Reviewer #2: Yes

5. Review Comments to the Author

Reviewer #2: The manuscript is well written and the research is conducted accordingly with methodological rigor.

There is no research or evidence of publication ethical issues in the this manuscript to the best of my understanding.

6. PLOS authors have the option to publish the peer review history of their article (what does this mean?). If published, this will include your full peer review and any attached files.

Do you want your identity to be public for this peer review? For information about this choice, including consent withdrawal, please see our Privacy Policy.

Reviewer #1: No

Reviewer #2: No

Thanks so much for the feedback. The manuscript has been revised in accordance to the PLOSONE style templates as recommended.

The write up has been edited as advised. The revised manuscript now appears as “This work was funded by the Swedish Research Council (external to my organization), (Grant 2019-04256), Makerere University (internal) and the MakRif Project (internal). There was no other additional external funding received for this study. The content is solely the responsibility of the authors and does not necessarily represent the official views of Swedish Research Council and Makerere University or the MakRif Project. The funders provided support in the form of tuition waiver and research expenses, but did not have any additional role in the study design, data collection and analysis, decision to publish, or preparation of the manuscript”.

Thanks so much for the caution. We have deleted the information on funding in the Acknowledgement statement. The write up now appears as “The lead author along with the co-authors extend their gratitude to the participants, doctoral committee and the research team that enable this study to become a reality. This study was undertaken as part of the lead author’s doctoral dissertation.

The amendment has been made in the write up as advised.

The write up has been edited to avail all the additional information. The write up now appears as “all data underlying the study’s findings have been availed in the manuscript. Additional supplementary information such as the consent forms, interview guide and administrative clearance forms have been availed as supplementary files”.

The reference list has been edited and reviewed as recommended.

---

## [Editor Report · Decision Letter 1]

Perceptions of Women and their Partners on Postabortion Intrauterine Contraception: A Qualitative study in central Uganda

PONE-D-24-58782R1

Dear Dr. Kayiga

We’re pleased to inform you that your manuscript has been judged scientifically suitable for publication and will be formally accepted for publication once it meets all outstanding technical requirements.

Kind regards,

Adnan Ahmad Khan, MBBS, MS.

Academic Editor

PLOS ONE
---

## [Editor Report · Acceptance letter]

PONE-D-24-58782R1

PLOS ONE

Dear Dr. Kayiga,

I'm pleased to inform you that your manuscript has been deemed suitable for publication in PLOS ONE. Congratulations! Your manuscript is now being handed over to our production team.

Kind regards,

on behalf of

Dr Adnan Ahmad Khan

Academic Editor

PLOS ONE